# Dietary Thiols: A Potential Supporting Strategy against Oxidative Stress in Heart Failure and Muscular Damage during Sports Activity

**DOI:** 10.3390/ijerph17249424

**Published:** 2020-12-16

**Authors:** Mariarita Brancaccio, Cristina Mennitti, Arturo Cesaro, Fabio Fimiani, Elisabetta Moscarella, Martina Caiazza, Felice Gragnano, Annaluisa Ranieri, Giovanni D’Alicandro, Nadia Tinto, Cristina Mazzaccara, Barbara Lombardo, Raffaela Pero, Giuseppe Limongelli, Giulia Frisso, Paolo Calabrò, Olga Scudiero

**Affiliations:** 1Department of Biology and Evolution of Marine Organisms, Stazione Zoologica Anton Dohrn, Villa Comunale, 80121 Naples, Italy; mariarita.brancaccio@szn.it; 2Department of Molecular Medicine and Medical Biotechnology, University of Naples Federico II, Via S. Pansini 5, 80131 Naples, Italy; cristinamennitti@libero.it (C.M.); nadia.tinto@unina.it (N.T.); cristina.mazzaccara@unina.it (C.M.); barbara.lombardo@unina.it (B.L.); pero@unina.it (R.P.); 3Department of Translational Medical Sciences, University of Campania “Luigi Vanvitelli”, 81100 Naples, Italy; arturocesaro@hotmail.it (A.C.); elisabetta.moscarella@unicampania.it (E.M.); gragnano.f@gmail.com (F.G.); limongelligiuseppe@libero.it (G.L.); 4Division of Clinical Cardiology, A.O.R.N. “Sant’Anna e San Sebastiano”, 81100 Caserta, Italy; 5Inherited and Rare Cardiovascular Diseases, Department of Translational Medical Sciences, University of Campania “Luigi Vanvitelli”, Monaldi Hospital, 81100 Naples, Italy; fimianifabio@hotmail.it (F.F.); martina.caiazza@yahoo.it (M.C.); 6Ceinge Biotecnologie Avanzate S. C. a R. L., 80131 Naples, Italy; ranieria@ceinge.unina.it; 7Department of Neuroscience and Rehabilitation, Center of Sports Medicine and Disability, AORN, Santobono-Pausillipon, 80122 Naples, Italy; ninodalicandro@libero.it; 8Task Force on Microbiome Studies, University of Naples Federico II, 80100 Naples, Italy

**Keywords:** sulfur-containing compounds, oxidative stress, sport activity, athletes, heart failure, muscle damage and diet supplementation

## Abstract

Moderate exercise combined with proper nutrition are considered protective factors against cardiovascular disease and musculoskeletal disorders. However, physical activity is known not only to have positive effects. In fact, the achievement of a good performance requires a very high oxygen consumption, which leads to the formation of oxygen free radicals, responsible for premature cell aging and diseases such as heart failure and muscle injury. In this scenario, a primary role is played by antioxidants, in particular by natural antioxidants that can be taken through the diet. Natural antioxidants are molecules capable of counteracting oxygen free radicals without causing cellular cytotoxicity. In recent years, therefore, research has conducted numerous studies on the identification of natural micronutrients, in order to prevent or mitigate oxidative stress induced by physical activity by helping to support conventional drug therapies against heart failure and muscle damage. The aim of this review is to have an overview of how controlled physical activity and a diet rich in antioxidants can represent a “natural cure” to prevent imbalances caused by free oxygen radicals in diseases such as heart failure and muscle damage. In particular, we will focus on sulfur-containing compounds that have the ability to protect the body from oxidative stress. We will mainly focus on six natural antioxidants: glutathione, taurine, lipoic acid, sulforaphane, garlic and methylsulfonylmethane.

## 1. Introduction

The production of free radicals is a physiological event and normally occurs in cellular biochemical reactions [1,2,3], especially in those that use oxygen to produce energy. The production of free radicals can also be due to external additional factors such as pollution, smoking, drugs, alcohol, ultraviolet rays and ionizing radiation, prolonged psychophysical stress and food additives [4] (Figure 1).

Physiologically, free radicals have a dual role, that is, they can be both useful and harmful to the human body [5]. In small quantities, free radicals have important functions in the maturation processes of cellular structures and are used as real weapons by phagocytes, cells of the immune system whose function is to physically destroy the pathogens that manage to enter our organism. Reactive oxygen species (ROS) and reactive nitrogen species (RNS) also participate in numerous cell signaling systems: for example, nitric oxide is an important messenger in the processes of blood flow regulation and in certain conditions it participates in the destruction of pathogens and cancer cells. of essential elements for growth, development and defense of the cell and the organism. It is their excess that can become a serious problem [6]. 

However, when an overload of free radicals cannot be gradually processed by the human body by the organs and physiological detoxification systems (especially expressed in the liver and kidney), an accumulation of free radicals occurs in the body. The consequence of this accumulation is a condition known as “oxidative stress”. The body produces free radicals during normal metabolic processes. However, the accumulation of these free radicals can damage cells, proteins and DNA, helping to cause “oxidative damage” (Figure 1) [7].

Oxidative damage is usually considered the starting point for the appearance of various diseases; in addition, it plays a key role in the development of premature aging, chronic and degenerative disorders such as arthritis and muscle damage [8], autoimmune disorders, cardiovascular [9,10] and neurodegenerative diseases, inflammation and cancer [10,11,12,13,14,15,16,17,18,19,20]. 

The free radicals that are involved in oxidative stress damage are divided into (ROS) and (RNS). Overproduction of ROS generates cellular genomic instability causing carcinogenesis by promoting aberrant cell proliferation and uncontrolled apoptosis [21,22]. In addition, oxidative stress is responsible for epigenetic alterations, causing the activation of protooncogens (hypomethylation of the promoter regions) and the silencing of tumor suppressor genes (hypermethylation of the promoter region) [23]. On the other hand, the increase in ROS at the protein level can cause post-translational changes. In fact, proteins are the target of choice for ROS [24]. Post-translational modifications occur mainly at the expense of the residues of cysteine, methionine and histidine. Therefore, oxidative stress can determine changes in the usual physiological pathways, affecting different organs, causing serious consequences on the different phases of human life and on the basis of the sex of the subject affected by this manifestation [25,26]. Among the organs most affected by ROS we find the muscle and the heart.

Furthermore, the mitochondria are particularly vulnerable to oxidative damage and they are the organelles responsible for generating energy to supply the muscles; consequently, the damage to the mitochondria can cause muscle damage. In fact, with the increase in oxidative stress, the membranes of the mitochondria are seriously compromised, generating a reduction in the biogenesis of the mitochondria [27,28].

Moreover, oxidative stress has been shown to play an important role in the pathophysiology of cardiac remodeling and heart failure (HF) [29]. In particular, ROS can compromise the contractile capacity of cardiac cells. In addition, they stimulate the proliferation of cardiac fibroblasts and activate the matrix metalloproteinases (MMP), leading to the remodeling of the extracellular matrix. These cellular events are involved in the development and progression of adaptive myocardial remodeling and failure.

In addition, heart failure and muscle damage are two phenomena that affect professional athletes. In particular, heart failure has an incidence of 1/100,000 athletes [30,31,32,33], while the incidence of muscle injuries in athletes is about 30–40% of the total injuries [34,35].

However, it is known that before heart failure and irreparable muscle damage, it is possible to undertake rehabilitation of the athlete, which involves both a delayed pharmacological therapy and a support therapy that sees the use of natural antioxidants as protagonists, accompanied by a period of moderate physical activity.

This review provides the “state of the art” to discuss the benefits of constant physical activity and the integration of antioxidant compounds through diet to protect the body from oxidative stress. The focus will be on six natural antioxidants: glutathione, taurine, lipoic acid, sulforaphane, garlic and methylsulfonylmethane.

We will describe their properties and activities as well as their benefits on physical activity and on the ability to prevent heart failure or muscle damage.

## 2. Mechanisms of Oxidation

The oxidation and reduction reactions in chemistry are called redox reactions; in this case, the oxidation of an element occurs when this chemical element undergoes an electron subtraction which results in an increase in its oxidation number; at the same time, this subtraction of electrons can take place by another element, which thus undergoes the complementary reduction process.

In a biological environment, electron acceptors are defined as oxidants and/or pro-oxidants agents; conversely, those who donate electrons are defined as reducing agents and/or antioxidants.

The redox balance of a cell and consequently of the human body is due to the balance between pro-oxidative and antioxidant agents, this balance is of fundamental importance for the modulation of the various cellular metabolic signaling processes [36]. 

Mitochondria are considered to be the cell’s powerhouse. Cellular respiration takes place within them, with which they are able to produce large amounts of energy in the form of Adenosine Triphosphate (ATP) molecules. Consequently, mitochondria require oxygen to produce energy [37]. During the process of oxidative phosphorylation, most of the oxygen is reduced to water, while a small percentage is not reduced, resulting in the production of intermediate metabolites known as ROS [37].

At the same time, when the reagents are derived from nitrogen, they are called reactive nitrogen species (RNS). The enzyme responsible for ROS production is nitrate reductase (NADH). Reactive species can be classified into two categories: free radicals (superoxide anion, nitric oxide or nitric dioxide radicals) and non-radical derivatives (singlet oxygen, ozone, hydrogen peroxide, peroxynitrite, hypochlorous acid, organic peroxides and aldehydes).

It is known that cells are continuously exposed to stimuli that can influence the correct cell balance, these stimuli can be both exogenous (external factors) and endogenous (internal factors). Endogenous stimuli are the most dangerous as they are constant, which is why it is essential to reduce exogenous stimuli, to allow cells and organs not to become fatigued.

ROS can damage both DNA and proteins [38,39,40]. In particular, ROS promote against DNA different dangerous action [38,39] such as: deoxyribose oxidation, strand breakage, removal of nucleotides, modification of bases and DNA-protein crosslinks; on the other hand, ROS also exhibits a dangerous action against protein [40], for example: increased susceptibility of the protein against proteolysis, site-specific modification of amino acids, alteration of the electrical charge and enzymatic inactivation. In addition, ROS can oxidize to polyunsaturated free fatty acids, starting lipoprotein oxidation, generating loss of fluidity and permeability of cell membranes. 

In addition, ROS can also affect muscle metabolism. In fact, muscle can be influenced by both endogenous and ROS-generating factors such as nicotinamide adenine dinucleotide phosphate (NADPH), oxidases (NOXs), phospholipase A2 (PLA2), xanthine oxidase (XO) and lipoxygenases.; both from exogenous factors such as intense and prolonged physical exercise such as the marathon.

In this case, intense exercise causes an increase in pro-inflammatory cytokines such as tumor necrosis factor (TNF)-α, interleukin (IL)-1β, IL-1 receptor antagonist (IL-1ra), TNF receptors (TNF-R), IL-8 and macrophage inflammatory proteins (MIP)-1. In addition, there is also an increase in Interleukin-6 (IL-6), which has a dual pro-inflammatory and anti-inflammatory role. As an anti-inflammatory agent, it is produced during skeletal muscle contractions after intense training sessions [41,42,43]. In fact, during exercise, IL-6 appears to be involved in the mobilization of extracellular substrates to increase the supply of nutrients to the muscle [44]. On the other hand, ROS can activate the nuclear factor (NF-κB), the latter plays a critical role in the mediation of immune, inflammatory responses and in the apoptotic process. The aberrant regulation of NF-κB is associated with a number of chronic conditions, including diabetes and atherosclerosis. Recently, it has been shown that myostatin, which is the limiting factor for muscle growth, can be influenced by NF-κB [44].

Another phenomenon that generates ROS is lipid peroxidation [45,46,47,48], which produces reactive aldehydes, such as malondialdehyde (MDA) and 4-hydroxynonenal (HNE), the second, also defined as “second messenger of free radicals” is the main product of lipid peroxidation due to its involvement in numerous biological activities [49,50].

However, the main source of ROS in the human body is the immune system [51], as the immune system together with its components is involved in the defense of the organism from external stimuli that can trigger the inflammatory process [52,53,54] or from internal stimuli of cell regeneration [55].

When cell damage occurs in a tissue, the immune system activates along with its cellular component which, in response to the insult, produces ROS, causing the onset of the inflammatory process [56]. 

Furthermore, ROS activate a wide variety of factors and kinases involved in both hypertrophy and apoptotic processes. In addition, they stimulate the proliferation of cardiac fibroblasts and activate the matrix metalloproteinases, indicating the remodeling of the extracellular matrix. These cellular events are involved in the development and progression of maladaptive remodeling and myocardial failure [29,30,31,32,33]. The accumulation of ROS is often also caused by dietary imbalances and incorrect lifestyles that can cause the production of pro-inflammatory hormones such as cortisol and insulin [57]. The increase in ROS can cause numerous breakdowns which, cause irreversible damage.

## 3. Antioxidant Role of Physical Exercise

The link between sport and oxidative stress is proportional to the intensity of physical activity. Physical exercise physiologically generates greater production of ROS, in relation to muscle work, especially if aerobic [58]. A correct program of physical activity or a rational muscle training generates a moderate and short-term increase in free radicals, which can activate molecular mechanisms useful to the cell to adapt and protect itself from states of oxidative stress, and to improve the immunological defenses of the organism (Figure 2) [59].

The increase in ROS, initially associated with the concept of oxidative stress and possible damage to muscle fibers, is instead recognized today as an important physiological mechanism for regulating cellular processes. It has been shown that ROS produced during moderate-intensity physical exercise are generally counterbalanced by the increased effectiveness of antioxidant systems [59,60].

Furthermore, ROS, in such circumstances, favor a positive regulation of gene transcription, which results in an increase in mitochondrial biogenesis and, consequently, an increase in the energy potentially developed by the muscle which assumes greater oxidizing capacity. For these reasons, when the muscle works it increases its ability to produce and use energy [59,60].

Moreover, ROS activate the pathways involved in enhancing the activity of antioxidant enzymes and induce the production of some cytokines capable of playing an anti-inflammatory role [61,62].

In addition, it has been shown that ROS produced during exercise promote the generation of enzymes involved in the repair of any damage suffered by DNA and are capable of depressing the expression of proteins responsible for cell death.

A man/woman, in his/her daily routine, can favor or delay the formation of free radicals and, consequently, the damage they produce.

In particular, the body has anti-oxidant defense systems that divide into antioxidant enzymes (superoxide dismutase, catalase and glutathione peroxidase, etc.) and non-enzymatic antioxidants (coenzyme Q10, glutathione, uric acid, lipoic acid, bilirubin, etc.). Among non-enzymatic antioxidant agents, coenzyme Q10 (CoQ10), also called ubiquinone, plays a fundamental role [63]. It is mainly expressed at the mitochondrial level, and plays an essential role in energy production. Its reduced form, ubiquinol, acts as an important antioxidant in the body. CoQ10 is synthesized endogenously and its absorption through nutrition is limited.

At the same time, adequate physical exercise alongside proper nutrition is essential for reducing oxidative stress, promoting the best state of health and preventing cellular aging.

In fact, the protection offered by antioxidants is variable, depends on multiple factors and requires close medical supervision: natural contributions of more antioxidants have shown modulating and strengthening effects of the physiological antioxidant barrier, while excessive doses of single substances have proved counterproductive, favoring the appearance of chronic degenerative diseases, possibly even cancers [64].

In a healthy person, in conditions of rest, the levels of free radicals are lower than those of an active person and tend to increase with physical exercise.

A healthy organism, properly trained and fed, carrying out muscular activity is however able to maintain the levels of ROS and RNS within normal ranges: this occurs because it is equipped with an adequate antioxidant barrier, part of food origin and part induced by the same radicals produced during the year [65].

However, when the individual undergoes extreme physical-sporting activities, whether competitive or non-competitive, high and persistent concentrations of ROS and RNS are generated, which can exceed the body’s antioxidant capacity. If these circumstances persist for a long time, harmful effects can be configured, which activate proteolytic processes, which trigger cellular degenerative mechanisms. These, in turn, in more or less long durations, can manifest themselves with the most various pathologies [66]. In this case, the strong link between sport and Oxidative Stress emerges.

## 4. Antioxidant: Sulfur-Containing Compounds

Antioxidant is any substance capable of interfering with the chemical oxidation reactions that give rise to oxygen or nitrogen free radicals or of neutralizing those already produced. This category includes very different compounds, naturally present in the body (endogenous) or taken with food (exogenous). The anti-oxidant, anti- inflammatory and anti-carcinogenic properties of many foods are linked precisely to their precious antioxidant content. McLeay and co-workers have shown the effectiveness of different natural products and phytochemicals present in food and used as food extracts (such as glutathione, taurine, lipoic acid, sulforaphane, garlic and methylsulphonylmethane) to prevent or reduce the progression of ROS accumulation at inside the human body [67]. In particular, the use of sulfur-containing antioxidants, capable of inducing a sensitive and/or substantial reversal of the damage caused by oxidative stress is increasingly used in the sports world (Table 1.) [67]. 

Thiols, molecules that contain a sulfhydryl (SH) side chain group, act as antioxidants, stabilizing free radicals by accepting their unpaired electron [68,69]. In fact, thiols play a crucial role in living organisms, as they contribute to maintaining the cellular redox homeostasis by regulating the oxidation-reduction potentials and thiol-disulfide protein ratios [70].

In particular, the oxide-sulfur activities are attributed to the reactivity that these sulfur-containing compounds have against ROS and NO, through the formation of radicals, preventing damage and nitrosothiols from oxidation, key mediators in the signaling of NO, with a physiological effect and therapeutic impact on many tissues and organs. Hence, the properties of the sulfydryl groups give greater importance to biological activities to natural products containing sulfur [68,69].

However, it is necessary to pay attention to the use of these compounds as food supplements, as an incorrect intake and therefore an excess of antioxidants can be negative for human health causing immunosuppression [71].

### 4.1. Glutathione

Among the molecules that play a pivotal role in protecting against oxidative stress there is glutathione (GSH).

Glutathione is a tripeptide made up of L-cysteine, L-glutamic acid, and glycine (Figure 3). It is an endogenous antioxidant, it protects cells by playing an oxygen radical scavenger role. A decrease in cellular GSH content increases ROS [72]. GSH is the most abundant thiol in the cell, it is synthesized inside the cell and partially secreted in the extracellular space along a concentration gradient. In addition, GSH has a variety of biological functions such as the regulation of numerous enzymatic activities, receptors, transcription factors, and finally transduction of the redox-sensitive signal, short-term conservation of cysteine, protein structure, cell growth, proliferation and programmed cell death [73]. Therefore, an adequate level of endogenous GSH is fundamental in maintaining redox balance within the body’s tissues with the ratio of reduced and oxidized glutathione (GSH/GSSG) which represents a primary indicator of cellular balance. A higher ratio of GSH to GSSG suggests a reductive environment where ROS levels are maintained at homeostatic levels, while a low GSH/GSSG ratio is indicative of oxidative stress [74]. It is the cysteine that gives GSH its antioxidant activity; since cysteine is also limiting its formation, dietary cysteine, or its amino acid precursor methionine, it is crucial for maintaining endogenous antioxidant defense. Various studies have shown that the integration of thiol donors has a positive effect on the up-regulation of GSH. For example, administration of glutathione or glutathione precursors such as cysteine-donor N-acetyl cysteine (NAC) in humans significantly increases GSH levels in blood [74,75,76,77] and muscle [78,79,80] preventing the accumulation of ROS.

### 4.2. Lipoic Acid

Lipoic acid (ALA) is a natural organosulfur compound (Figure 3), introduced through the diet by consuming broccoli, tomatoes, spinach, salads, cabbage, peas, brewer’s yeast, brown rice and meat [81]. In addition, ALA can be both fat-soluble and water-soluble, therefore it acts and performs its function on a considerable number of free radicals, both inside and outside the cell. The generation of reactive nitrogen/oxygen species (RNS/ROS) represents an important mechanism in erythropoietin (EPO) expression and skeletal muscle adaptation to physical and metabolic stress. Morawin et al. have shown that the generation of RNS/ROS can be modulated by controlled physical exercise and by taking food supplements such as ALA which has both an anti- and pro-oxidative action [82]. The study was planned to highlight how the intake of ALA in combination with physical exercise modulated the hematological response. Sixteen healthy young males participated in the randomized, placebo-controlled study. The exercise test involved a 90-min run followed by a 15-min eccentric phase at 65% VO2max (−10% gradient). The physical exercise thus carried out caused an increase in serum of nitric oxide (NO) serum, hydrogen peroxide (H_2_O_2_) and pro-oxidative products such as 8-isoprostane (8-iso), lipid peroxides (LPO) and carbonyl proteins (PC). Taking ALA (Thiogamma: 1200 mg daily for 10 days before exercise) caused a 2-fold increase in serum H_2_O_2_ concentration before exercise but prevented the generation of nitrogen oxide (NO), 8-iso, LPO and PC at 20 min, 24 h and 48 h after exercise. In addition, the intake of ALA also elevated the serum level of EPO, which is strongly correlated with the NO/H_2_O_2_ ratio. Total serum creatine kinase (CK) activity, as a marker of muscle damage, peaked 24 h after exercise (placebo 732 ± 207 IU·L^−1^, ALA 481 ± 103 IU·L^−1^) and correlated with the increase in EPO in the group of subjects in which ALA was taken. In conclusion, the intake of ALA acid modulates the generation of RNS/ROS, improves the release of EPO and reduces muscle damage after performing an intense exercise.

### 4.3. Taurine

Taurine is a substance (to be precise, it is an aminoethanesulfonic acid) (Figure 3) that is produced in the human body by the oxidative catabolism of cysteine or by oxidation of hypotaurine. Taurine is found in large quantities in the brain, retina, heart and platelets. In the form of taurocholic acid and taurodeoxycholic acid it enters the constitution of bile acids; in the central nervous system it is a neurotransmitter with inhibitory action. The best food sources are meat and fish. Taurine-based supplements have also been developed due to several therapeutic applications [83]. In adults, taurine can be synthesized in the liver starting from cysteine and methionine in the presence of sufficient amounts of vitamin B6. Taurine is involved in many cellular biological activities such as bile acid conjugation, cell membrane stabilization, and calcium signaling.

De Carvalho and co-authors have shown that the administration of taurine through the diet does not increase the athlete’s aerobic parameters [84], although there is an increase in the plasma levels of taurine and a concomitant decrease in the markers of oxidative stress, suggesting that the taurine prevents the accumulation of ROS/RNS in athletes. The study was conducted in a group of 10 male triathletes, aged 30.9 ± 1.3 years, height 1.79 ± 0.01 m and body weight 77.45 ± 2.4 kg. Three grams of taurine and 400 mL of chocolate milk, or a placebo (chocolate milk) was ingested post exercise for 8 weeks. Oxidative stress marker levels, and 24 h urinary nitrogen, creatinine, and urea excretion were measured before and after 8 weeks.

### 4.4. Garlic

Garlic (Allium sativum), having a huge number of bioactive properties mainly due to its content in organosulfur compounds (Figure 3), which are able to regulate the expression of a wide variety of genes, including the inducible nitric oxide synthase (iNOS), with a strong impact on human health [85]. The molecule that allows garlic to exert this property is alliin (Figure 3).

Ince and colleagues [86] shed light on how a single administration of garlic to a group of college endurance athletes can influence aerobic presentation. In the present case, 900-mg dose of dried garlic powder or placebo was administered randomly in a double-blind cross-over fashion. Five hours after ingestion of the tablets, the subjects underwent an incremental treadmill running test according to the Bruce protocol until subjective exhaustion. During the test, blood pressure and heart rate were monitored at 3 min intervals. In this scenario, there was a significant increase in maximum oxygen consumption (VO2max) improving the performance of the athletes.

### 4.5. Sulforaphane

Sulforaphane (SFN) is a dietary isothiocyanate (Figure 3) obtained by the enzymatic processing of glucopharanin, a 4-methylsulfinylbutyl glucosinolate, present in cruciferous vegetables, such as broccoli and cabbage. Recent studies have shown that sulforaphane treatment protects skeletal muscle against damage induced by exhaustive exercise in rats. In this case, Malaguti and co-workers [87] used a group of Male Wistar rats who received SFN (25 mg/kg body wt ip) for 3 days before undergoing an acute exhaustive exercise protocol in a treadmill (+7% slope and 24 m/min). Acute exercise resulted in a significant increase in plasma lactate dehydrogenase (LDH) and creatine phosphokinase (CPK) activities. It also resulted in a significant increase in thiobarbituric acid-reactive substances, in a significant decrease in tissue total antioxidant capacity, and in a significant decrease in NAD(P)H: quinone oxidoreductase 1 (NQO1) expression and activity in vastus lateralis muscle. SFN treatment increased muscle NQO1 and caused an upregulation of phase II enzymes. The up regulation of the detoxification enzymes was accompanied by a significant increase in nuclear erythroid 2 p45-related factor 2 expression and correlated with a significant increase in total antioxidant capacity and a decrease in plasma LDH and CPK activities. These data suggest that the administration of SFN plays an indirect antioxidant role on the muscle by preventing muscle damage which intense exercise can cause.

### 4.6. Methylsulfonylmethane

Methylsulfonylmethane (MSM), also known as dimethyl sulfone or methyl sulfone, is a natural organosulfur compound present in foods like fruits, vegetables, grains, coffee, beer, tea and cow’s milk [88] (Figure 3). Its main application is in the form of high-temperature, polar, aprotic, commercial solvent, as is its parent compound, dimethyl sulfoxide (DMSO) [89]. MSM exerts the anti-inflammatory activity through an inhibitory effect of MSM on NF-kB, which results in a downregulation of mRNA for interleukin (IL)-1, IL-6, and tumor necrosis factor-α (TNF-α) in vitro [90]. MSM can also diminish the expression of inducible nitric oxide synthase (iNOS) and cyclooxygenase-2 (COX-2) through the suppression of NF-κB and reducing the production of vasodilators such as nitric oxide (NO). The latter also regulates mast cell activation, therefore MSM indirectly could have an inhibitory effect on mast cell mediation of inflammation [91]. MSM modulates the activation of at least four types of transcription factors: NF-κB, p53, and nuclear factor (erythroid-derived 2)-like 2 (Nrf2). By mediating these transcription factors, MSM can regulate the balance of ROS and antioxidant enzymes [92]. Moreover, MSM also has effects on the immune system. It has been shown that MSM can reduce IL-6 in vitro, mitigating the deleterious effects of IL-6 in the maintenance of chronic inflammation [93].

A study conducted on 24 men accustomed to exercise showed that administering MSM 3.0 gm/d or placebo for 14 days with a 17-day washout in between significantly reduced IL-6 levels after one session of intense training. Consequently, the use of this antioxidant agent can certainly relieve post-exercise muscle fatigue/damage [94].

## 5. Use of Sulfur-Containing Compounds in the Treatment of Heart Failure in Professional Athletes

The heart of professional athletes usually undergoes modifications due to sustained physical activity, with a benign increase in cardiac mass and specific circulatory and cardiac morphological alterations, that represents a physiological adaptation to training [95,96,97,98]. However, the risk of heart failure (HF) due to unrecognized symptomatic or asymptomatic heart chambers remodeling in athletes increases with age and drug abuse [99,100]. The main cause of HF in athletes is dilated cardiomyopathy, and in young athletes, myocarditis is the most frequent aethyology of acquired dilated cardiomyopathy (Figure 4) [101,102,103]. Despite the fact that moderate sports activity seems to have an antioxidant role, the activity of professional athletes seems to produce an excessive amount of ROS [104]. In fact, elite athletes seem to be exposed to increased oxidative stress as a result of intensive training, and an excess production of ROS has been shown to play an important role in the pathophysiology of cardiac remodeling and HF [28,101,102,103,104]. ROSs stimulate cardiac fibroblast proliferation and activate the matrix metalloproteinases, leading to the extracellular matrix remodeling [105,106]. In this context, there is a growing interest in the use of antioxidant supplements. In fact, the appropriate use of antioxidants could play a dual role: (i) reducing oxidative stress in athletes and slowing the progression to HF; (ii) providing support in the therapy of HF patients. In particular, among antioxidants, sulfur-containing compounds seem to play a crucial role in reducing oxidative stress (Figure 4) [107,108,109]. 

N-acetylcysteine is the oral glutathione precursor, and its supplementation seems to increase exercise performance and reduces oxidative stress in individuals with low levels of glutathione [110]. Cardiac and systemic glutathione deficit has shown, in animal models, cardiac remodeling and heart failure progression. Damy et al. have compared cardiac and blood glutathione concentrations in patients with different functional classes [111]. They showed that in atrial tissue, glutathione was depleted (−58%) in New York Heart Association (NYHA) class IV patients compared to NYHA class I patients (*p*  =  0.002). Compared to controls, blood levels of glutathione were decreased by 40% in symptomatic patients of NYHA class II to IV (*p* < 0.0001). Adamy et al. showed that in a rat model of HF post myocardial infarction there was a glutathione deficiency compared with control animals [112]. One-month oral N-acetylcysteine supplementation normalized glutathione levels improved left ventricle contractile function and reduced adverse left ventricle remodeling, contrasting with the decline of heart function [112]. In fact, data from other animal studies confirm that N-acetylcysteine administration can reduce cardiac fibrosis and remodeling in heart failure hypothetically by reducing oxidative stress [108,109,110,111,112,113]. Despite available literature data, the role of glutathione supplement, in both normal and pathological cardiovascular conditions, still remains unclear. 

Taurine has antioxidant activity and modulation of calcium homeostasis. In athletes, taurine can improve calcium transport to myofibrillar contractile proteins, optimizing skeletal muscle function, resulting in better performance [114]. Zhang et al. showed that 6 g/day of taurine supplementation for seven days increased the time to exhaustion, maximum workload, and maximal oxygen uptake on a cycle ergometer, reducing the oxidative stress markers [115]. In fact, the use of taurine supplementation may oppose the possible overproduction of ROS, avoiding their effects on cardiac remodeling [116,117]. In heart failure patients, taurine supplementation seems to be useful to improving some selected hemodynamic parameters [118]. The cardiac taurine concentration is about 20 mM, about 100 times higher than its blood levels and its antioxidant beneficial effects are achieved in patients with HF through various mechanisms [119]. Taurine is a scavenger of hypochlorous acid, a highly toxic oxidant, which is produced by neutrophils; it prevents auto-oxidation of adrenaline to adrenochrome, an oxidant involved in HF progression [119,120]; and seems to counteract the neurohormonal response triggered by the increase in angiotensin II and the activation of the sympathetic nervous system. Taurine could protect against angiotensin II-mediated cardiac fibrosis, because it showed that avoids angiotensin II-enhanced cell proliferation [121] and expression of c-fos and c-jun early response genes in cultured cardiac fibroblasts [122]. 

Lipoic Acid has a protective effect on the cardiovascular system, particularly by acting on lipid profile and arterial blood pressure [123,124]. In athletes, it can extend workout time and improve muscle strength. Lipoic Acid helps avoid or repair muscle damage in both endurance and resistance exercise [125]. In HF patients, cardiomyocytes contrast the oxidative stress by increasing antioxidant system activity and increasing energy spending. Lipoic Acid can act as a cofactor for enzymatic reactions into the mitochondria and can increase mitochondrial function by conserving cellular energy [126,127]. Thus, Lipoic Acid can influence mitochondrial antioxidant status, counteract ROS, and mitigate mitochondrial damage caused by oxidative stress and the aging process. Despite promising premises, there is no major evidence in the literature on lipoic acid supplementation in HF patients. 

Sulforaphane, represents one of the most important isothiocyanates in the human diet, present in cruciferous vegetables, with antioxidant activities in different tissues [128,129]. In fact, sulforaphane has shown, in a rat model, antioxidant power in skeletal muscle modulating muscle redox environment, preventing muscle damage due to exhaustive exercise [87]. Ma et al. have shown that in a rabbit model of HF, treatment with sulforaphane improved cardiac function and remodeling by inhibiting oxidative stress and inflammation [130]. Sulforaphane exerts its protective action on the cardiovascular system by the activation of NF-E2-related factor 2 (Nrf2), a transcription factor that serves as a defense mechanism against oxidative stress by inducing more than a hundred cytoprotective proteins [131] and prevents angiotensin II-induced cardiomyopathy by Nrf2 via stimulating the Akt/GSK-3ß/Fyn pathway [132].

Furthermore, garlic and ergothioneine, which have recognized antioxidant properties, may have a potential protective role in patients with HF [133,134,135,136,137].

Although there is increasing evidence of the antioxidant effects of the above mentioned sulfur-containing compounds, it should be pointed out that while in athletes these elements could prevent and reduce oxidative stress, in patients with HF these can only be a support to the pharmacological therapy recommended by current guidelines [138] which remains the fundamental cornerstone of HF management.

## 6. Prevention of Muscle Damage by Sulfur-Containing Compounds

Physical exercise has been shown to generate physiologically an increase in ROS and RNS production but (Figure 5), when the athlete undergoes strenuous physical activity, the rate of reactive species is so elevated that it exceeds the body’s antioxidant capacity [139]. An excessive production of ROS during strenuous exercise also can have negative consequences on contractility and promote the onset of fatigue [140].

Moreover, the increase in ROS production often causes alterations of various biological substances, including lipids, proteins and nucleic acids, and the derivatives of these oxidatively-damaged substances can be used to obtain information on ROS production in various pathophysiological conditions. Most commonly markers of oxidative damage are lipid peroxidation, protein carbonyls and the changes in the levels of antioxidant molecules [141]. 

It has been suggested that there is a possible relationship between ROS production and exercise-induced muscle damage, and the mechanisms underlaying are not clear yet. Maughan et al. have demonstrated that an increase in lipid peroxidation products occurs in plasma after a downhill run [142]. Reznick et al. have reported that a single bout of exercise was responsible for an increase in protein-bound carbonyl content in the rat skeletal muscle [143]. Fielding et al. have shown that during the process of inflammation, the increase in immigrating neutrophil and their resulting release of ROS in excess can play an important role in this relationship [144]. 

Because ROS are involved in detrimental cellular processes, Morillas-Ruiz and collaborators have investigated the potential beneficial effects of antioxidant consumption [138], but their usage among athletes has given rise to controversy [145,146]. The differences in results are due to several factors such as methods, the antioxidant tested or the nutritional status of the participant in the test [147]. Indeed, Moini et al. have shown that the supplementation avoids the decline of other antioxidants such as glutathione and antioxidant vitamins, improves glucose metabolism and reduces exercise-induces oxidative damages in several tissues [148]. 

It has been shown that three-day administration with N-acetylcysteine or α-lipoic acid improves total antioxidant status and confirmed the significant antioxidant action through the reduction in lipid peroxidation and protein carbonylation, whereas daily administration of taurine did not influence plasma pro-antioxidant status in healthy men after performing a single muscle-damaging exercise [149].

On the other hand, the usage of N-acetylcysteine and α-lipoic acid can determine a temporary improvement of lipid peroxidation, inhibition of glycogen production and mitochondrial damage [150,151]. 

Some researchers have investigated the usage of methylsulfonylmethane (MSM), a sulfur-containing compound with low toxicity and contained in a wide variety of human foods including fruits, vegetables, and beverages [152,153]. MSM is used in the treatment of seasonal allergic rhinitis [154], autoimmune disease [155], and produces anti-inflammatory and antioxidant effects [155,156,157]. It has been demonstrated that a single dose oral supplementation of MSM reduces protein carbonylation and may increase plasma total antioxidant capacity, but it is not capable of increasing GSH levels in the plasma [139].

In conclusion, further studies are needed to establish whether the usage of sulfur-containing compounds can prevent oxidative damage due to exercise and, therefore, protect the athlete’s health (Figure 5).

## 7. Conclusions and Future Prospects

In recent years, the necessity of new and sustainable treatments for human pathologies has increased. In this scenario, research to identify new active compounds from natural sources has significantly been stimulated. Indeed, the development of new beneficial products, overcoming the side effects of common medicines, has attracted great interest in the pharmaceutical and nutraceutical sectors [158] for the development of new natural products from poorly exploited sources. A good deal of attention has been given the use of sulfur-containing compounds as alternative biomolecules in support of traditional medicine in common human disorders [159,160,161,162,163,164]. At the same time, moderate and controlled physical activity has always been seen as a support for the prevention of degenerative, cardiovascular and muscular diseases; as well as to prevent premature aging [57]. 

In conclusion, the data collected to date suggest that a dietary supplement with sulfur-containing compounds [64,67,165], having high antioxidative capacities and moderate and controlled physical activity, could represent a valid weapon against oxidative stress, capable of reducing the accumulation of ROS and/or RNS by alleviating and preventing the appearance of serious pathophysiological disorders (Figure 6). A healthy lifestyle is necessary to counteract the appearance of premature cell aging, heart disease and muscle damage, which is why it is recommended at any age.

## Figures and Tables

**Figure 1 ijerph-17-09424-f001:**
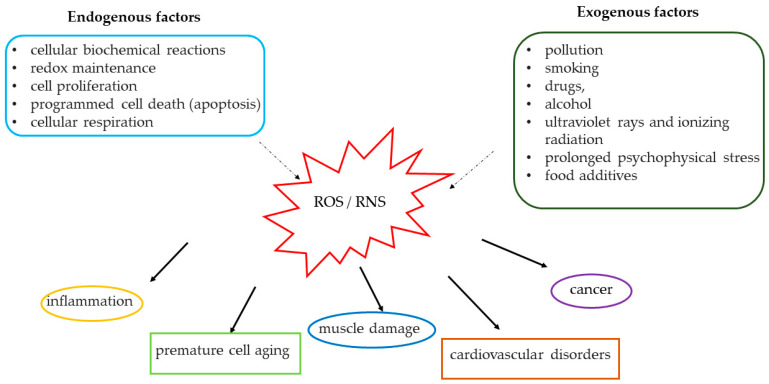
Causes and effects induced by the increase in reactive oxygen species (ROS) and reactive nitrogen species (RNS).

**Figure 2 ijerph-17-09424-f002:**
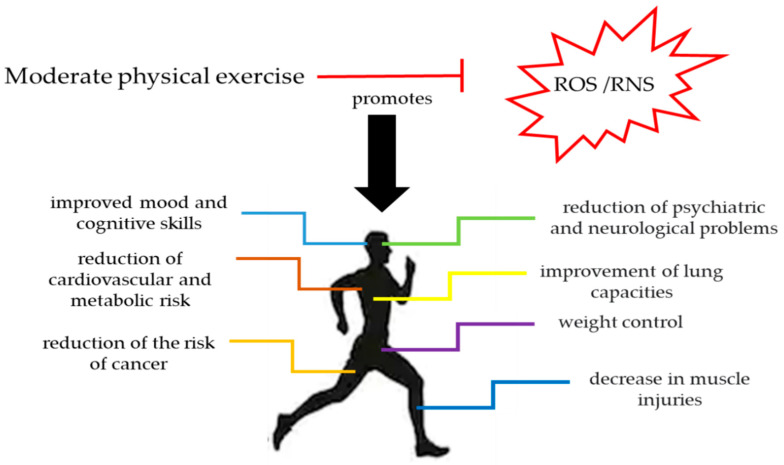
The benefits of moderate physical exercise.

**Figure 3 ijerph-17-09424-f003:**
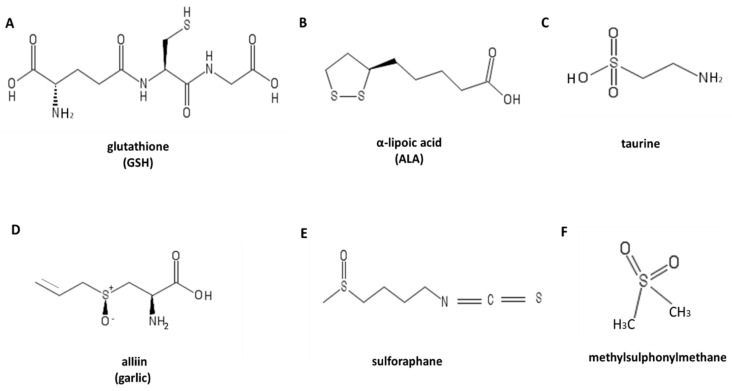
Chemical structures of sulfur-containing antioxidants with therapeutic potential against oxidative stress.

**Figure 4 ijerph-17-09424-f004:**
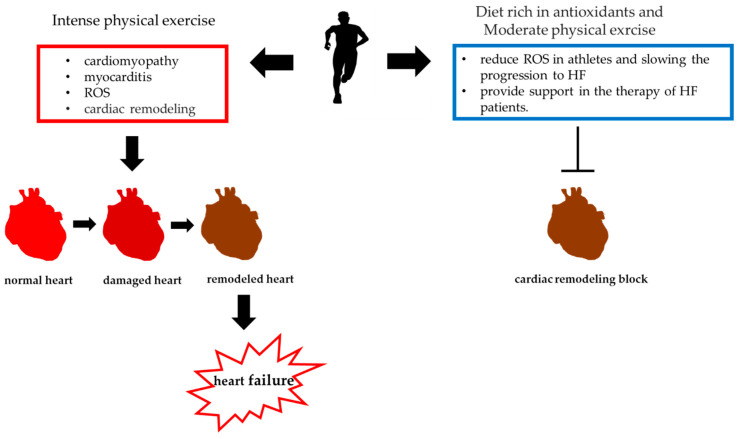
The role of intense physical exercise in heart failure.

**Figure 5 ijerph-17-09424-f005:**
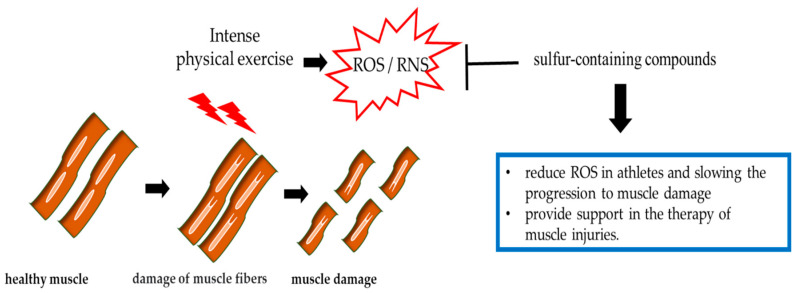
The effects of exercise-induced ROS/RNS on muscle damage.

**Figure 6 ijerph-17-09424-f006:**
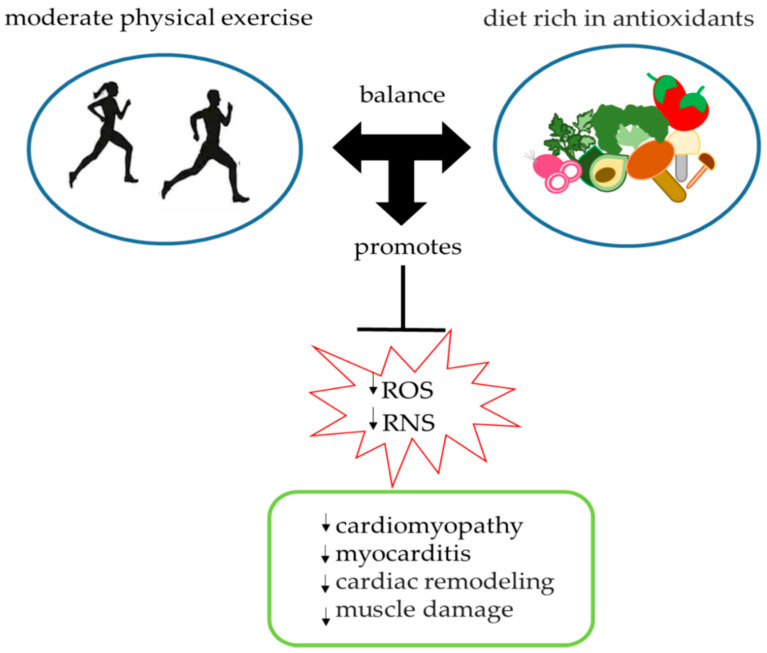
Correlation between the intake of foods rich in sulfur compounds and physical activity.

**Table 1 ijerph-17-09424-t001:** Therapeutic efficacy of sulfur-containing compounds in sports activity.

Supplement	Method of Administration	Type of Athletes	Beneficial Action	References
GSH	2 week oral GSH supplementation (1 g/day)	cyclists	reduction in muscle fatigue	[72]
ALA	1200 mg daily for 10 days before exercise	males subjected to physical exertion caused by intense running	reduces RNS/ROS formation, improves the release of EPO and reduces muscle damage	[76]
Taurina	3 g/day of taurine for 8 weeks	male triathletes	reduces RNS/ROS formation	[80]
Garlic	900-mg dose of dried garlic single administration	college endurance athletes	improving the performance of the athletes	[82]
SFN	25 mg/kg body wt ip	group of Male Wistar rats	preventing muscle damage	[83]
MSM	3.0 mg/day for 14 days	males subjected to physical exertion	reduction in muscle fatigue	[90]

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
