# Peer review of "Dietary Thiols: A Potential Supporting Strategy against Oxidative Stress in Heart Failure and Muscular Damage during Sports Activity"

_ijerph, 2020, doi:10.3390/ijerph17249424_

Round 1

Reviewer 1 Report

The review under the title "Dietary thiols: a potential supporting strategy in the treatment of cardiovascular and muscular disorders in sports activity" is dealing with an interesting aspect of sports nutrition. It is well written and well organised but there are some comments that need to addressed

  1. The title needs reconsideration. I believe that you are examining the thiol content in the prevention of oxidative stress and CVD, not in the treatment. So please restate
  2.  Please improve the quality of all the figures. They are too simplistic and naive. Figure 6 is not focused on the topic, it is a more general figure and it should be ommited. 
  3. Please explain what body wt ip stands for 
  4. Provide an explanation of the abbreviations the first time mentioned. 

Author Response

Dear Editor, 

thank you for the Reviewer’s Report about our manuscript entitled “Dietary thiols: a potential supporting strategy in the treatment of cardiovascular and muscular disorders in sports activity”, submitted to International Journal of Environmental Research and Public Health (IJERPH). We have appreciated the comments received by the Reviewers and yourself and have carefully considered them in preparing a new version of the manuscript.

A point-by-point response to the comments is attached below.

We believe that the manuscript is now significantly improved thanks to the Reviewer’s inputs.

We hope that the new version of the paper deserve publication on the Journal of Environmental Research and Public Health.

Best regards,

Prof. Dr. Olga Scudiero

Point-by-point response.

Reviewer 1

The review under the title "Dietary thiols: a potential supporting strategy in the treatment of cardiovascular and muscular disorders in sports activity" is dealing with an interesting aspect of sports nutrition. It is well written and well organised but there are some comments that need to addressed

  1. The title needs reconsideration. I believe that you are examining the thiol content in the prevention of oxidative stress and CVD, not in the treatment. So please restate
  2.  Please improve the quality of all the figures. They are too simplistic and naive. Figure 6 is not focused on the topic, it is a more general figure and it should be ommited. 
  3. Please explain what body wt ip stands for 
  4. Provide an explanation of the abbreviations the first time mentioned. 

Respone:

Thanks for the comments.

  1. In order to satisfy your request we modified the title. The new title is highlighted in yellow.
  2. We have modified the figures, in particular those where the legends are highlighted in yellow, to make them more explanatory in the eyes of the reader and to improve the quality of the manuscript.
  3. Body wt ip stands for body weight intraperitoneal and specifies that the indicated dose was administered per kg bodyweight and in an intraperitoneal injection
  4. In order to satisfy your request, we have added the missing abbreviations in the manuscript. They are highlighted in yellow

Reviewer 2

The manuscript entitled “Dietary thiols: a potential supporting strategy in the treatment of cardiovascular and muscular disorders in sports activity” by Brancaccio et al. is a narrative review that would like to point out how healthy lifestyle habits (through a combination of moderate physical exercise and balanced diet) could prevent free radical-mediated organ impairments (mainly heart and muscles). A special focus is put on natural sulfur-containing compounds and their antioxidative properties.

Despite the importance of the issue raised by the authors, the manuscript is really disappointing and many concerns make its publication at moment questionable.

-The title and the abstract do not match properly the content of the review.

In the title the focus seems to be on the potential benefits of dietary thiols in cardiovascular and muscular disorders (intended as pathologies) in sport. On the other hand, the abstract is focused on the importance of moderate physical exercise and a diet rich in antioxidants (sulfur-containing compounds) to counteract the harmful effects of oxygen free radicals; there are no clear statements regarding “cardiovascular and muscular disorders in sport”. Then, in the review there is a section on heart failure in professional athletes and one on free radical-mediated muscle damage, but not on muscular disorders as stated in the title.

-The general reading is unpleasant, above all the first three sections (Introduction, Mechanisms of oxidation, Antioxidant role of physical exercise). These sections suffer of a highly messy structure, where many provided notions are too basic and sloppy.

Respone:

Thanks for the comments.

  1. In order to satisfy your request, we have changed the title and rearranged the abstract so that there is no discrepancy.
  2. In the section entitled: Use of sulfur-containing compounds in the treatment of heart failure in professional athletes, clear reference is made to how heart attack can affect young athletes, the causes as well as possible treatments are specified. Finally, there is a reflection on how the use of natural antioxidants can support the usual pharmacological therapies and the prevention of HF caused by an accumulation of ROS. Also in section 6 entitled: Prevention of muscle damage by sulfur-containing compounds; lines 401 to 411 describe how intense physical activity causes an increase in ROS, which in turn causes muscle damage. The second part describes the use of natural molecules as a possible support therapy. In this way, both the title and the subsections do not present any contradictions.
  3. the first three paragraphs are intended to introduce the reader to oxidative stress and how this phenomenon causes damage to the athlete.at the same time, the paragraph emphasizes how moderate and controlled physical activity is also recommended in neurodegenerative pathologies, as well as in recovery from any trauma.In a review it seems right to make a summary of what is known both on oxidative stress and on physical activity.

-The 1st section

In the introduction (line 53) is cited the dual role of free radicals; nevertheless, the authors do not describe the “useful” one (neither in any other section of the manuscript). Moreover, there is no recall to the physical exercise, but for in the final paragraph where the aims of the review are reported.

Response:

thanks for the suggestion.

From lines 60 to 67 we have specified the beneficial effects of ROS. From lines 101 to 107 we have specified how physical activity and natural antioxidants can support heart damage and muscle damage. Finally, from lines 108 to 113 we have underlined the aims of the review.

-The 2nd section

As for the second section, I don’t agree with the sentence (lines 105-106) “a small percentage of oxygen that is not used as an energy source forms intermediate reactions known as ROS”: oxygen is not an energy source itself and the word “reactions” is a completely wrong concept here. The sentence (line 116) “ROS can damage both DNA and proteins” is a standalone statement: no other explanation is provided; while it is understandable that the focus of this review is different, this sentence needs to be commented. The sentence (line 121) “thiol groups….are essential for cysteine residues belonging to numerous key proteins such as kinases and phosphatases” contains another wrong concept (the -CH2SH side chain DEFINES the Cys amino acids!!) and another careless statement (Cys amino acids play key structural as well as functional roles in so many different classes of proteins, and kinases and phosphatases are not representative of the whole set). Again, the concept explained at lines 142-147, supported by different autocitations of the authors, is not explained with the molecular details that the reader would expect in a section called “Mechanism of oxidation”.

Respone:

thanks for the suggestion.

The points indicated have been revised and the bibliography has been updated.

-The 3rd section

This is the most scarce section within the manuscript. Too many concepts left without proper references, unusual statements (Physical activity is defined as any body movement involving smooth muscle!!), confounding paragraphs (check lines 170-174 compared to line 127-129) (check lines 138-141 compared to line 116-117: are ROS able to oxidize PUFA or is lipid peroxidation able to generate ROS? The subject is complex and a clarification is needed to avoid misunderstanding). Moreover, the authors stress a lot the importance of moderate physical exercise compared to intense physical exercise: the reader would benefit from an explanation that clarify these concepts. In addition, the strong interplay between ROS production and physical exercise should be provided here.

Respone:

thanks for the suggestion.

Paragraph 3 has been completely revised.

-The 4th section

The paragraph at lines 206-210 is not clear. Please rephrase the concepts.

Respone:

thanks for the suggestion.

The point indicated have been revised.

The parts in yellow highlight the changes made.

All the subsections related to the single sulfur-containing compounds reports the results of at least one previous study where the specific compound was tested as supplementation in athletes. The same example is not found in the Glutathione subsection: are there no cases?

Respone:

In section 4.1 we reported studies in which the precursor of GSH called NAC is administered and used to increase the availability of GSH. We have also added a new reference where GSH is administered directly (Krzysztof Grucza et al. 2018)

In addition, are any side effects known for such supplementations?

Respone:

In order to satisfy your request, in paragraph four we have added a consideration to line 251-253, and a reference number 67.

An exhaustive table reporting the type of supplementation, the method of assumption, the type of athletes and their physical exercise program, and the results obtained in terms of physical fitness and antioxidant effects could be useful to understand the importance of the issue discussed in the manuscript.

Respone:

thanks for the suggestion.

We have added a table to summarize the cited studies.

Please check paragraph at line 272-274: it is incomplete.

Respone:

thanks for the suggestion.

We have changed the sentence, now line The parts in yellow highlight the changes made.

-The 5th section

Here there is again a short description of the sulfur-containing compounds, described in the previous section. A more structured frame of the manuscript should be organized.

Please specify the acronym NYHA.

Response:

Dear Reviewer, thanks for your comment.

The manuscript has been revised and restructured removing the redundancy of the sulfur-containing

compounds description; so, the text in section 5 has been modified.

In particular, the changes are highlighted in yellow.

-The 6th section

Here a different sulfur-containing compound (methylsulfonylmethane, MSM) is discussed. Should it be included in the previous sections???

Thanks to the reviewer 2 for the helpful comment.

Response:

As you suggest, we have added a paragraph in the section 4.

-The 7th section

In my opinion the statement at line 439 (sulfur-containing compounds represent a useful natural therapy against oxidative stress) is in contrast with the paragraph at lines 422-424.

Respone:

thanks for the comments.

In the paragraph 7, there are no contradictions; in fact, in line 422-424 now become 499-502, we define sulphur-containing compounds as a valid alternative therapy / support to traditional medicine; following line 504 now line 506, we reiterate the anti-oxidant power of the aforementioned natural compounds.

The mentioned parts are highlighted in yellow.

-References

The authors adopted an odd and random system for the references reported in the manuscript: some of them are not enough appropriate (for example, references 30-32, or 54), others are completely lacking (lines 126, 129, 136, 156, 161, 164, 175-191).

Respone:

thanks for the comments.

The references have been modified, those that the reviewers considered inadequate have been eliminated and have been replaced.

-Figures

Not all figures are instrumental to the manuscript. Some of them could be merged.

Please note that Figure 1 contains two acronyms that are not specified before the recall of the figure in the text (lines 48-52) neither in the corresponding figure legend. In general figure legends are not exhaustive (above all those of figure 4 and 5)

Respone:

thanks for the comments.

The figures with the legends highlighted in yellow have been changed.

In addition, figure 1 contains the acronyms of ROS and RNS which have been specified in section 1 line 78-79 (highlighted in yellow)

Throughout the manuscript there are a lot of English errors and mistyping. The authors should consider a major English revision of the manuscript. A more structured frame of the manuscript should be organized, above all in the first sections.

Respone:

thanks for the comments.

Thanks to the suggestions of the reviewers we have revisited some parts of the sections in order to make the reading more fluid, we have also updated both the references and the figures to make them suitable for the manuscript.

Finally, we did English editing.

Check also the acronyms: sometimes are used, sometimes are not, sometimes are specified again in a new section. Please verify the NAPDH acronym: at line 108 it refers to an enzyme, at line 125 to the cofactor, and at line 291 is written differently!

Respone:

thanks for the comments.

We have corrected the error and also reported the correct wording in the abbreviation.

Reviewer 3

The authors provide an interesting review about the potential uses of sulphur compound supplementation during sports acitivity.

Response:

Thanks to the reviewer 3 for the helpful comments.

The following revisions are necessary:

Line 57: clarify

Response:

In order to satisfy your request, we have modified text, line 57

All figures: please improve graphical resolution

Response:

In order to satisfy your request, we have modified the figures

Line 109 and 110: delete „such as“

Response:

In order to satisfy your request, we have modified text, lines now 108,109

Line 118 and throughout: please lower-case the 2s in H2O2

In order to satisfy your request, we have modified text. The parts in yellow highlight the changes made.

Line 134: tsthe?

It is a typing error.

In order to satisfy your request, we have modified text. The parts in yellow highlight the changes made.

Line 199: „several studies“? Which ones? Were these in vitro or in vivo studies? Probably the clinical evidence for this association is insufficient?

In order to satisfy your request , we reorganized the period, citing the author and the work.

Line 217: clarify

Respone:

We have clarified the definition assumed in line 217 now line 229. The added part is underlined in yellow.

Line 232: should read lipoic acid

Response:

In order to satisfy your request, we have modified the text. The parts in yellow highlight the changes made.

Line 257: „it“ missing. Clarify meaning of second sentence. Assume is not corrrect terminology

Response:

The average daily synthesis of taurine in adults ranges between 0.4-1.0 mmol (50-125 mg)1 ; under stress the synthesis capacity may be impaired; therewith some authors consider taurine as a conditionally essential amino acid. [R. Lourenço M. E. Camilo. Taurine: a conditionally essential amino acid in humans? An overview in health and disease. Nutr. Hosp. (2002) XVII (6) 262-270].

Moreover, we have modified the text, now line 256.

Line 284: „such as broccoli“

Response:

In order to satisfy your request, we have modified the text. The parts in yellow highlight the changes made.

Figure 3: is there a structure for garlic? This should be clarified

Response:

The structure reported in Figure 3 belongs to Alliin, a sulfur compound found in garlic. We have modified the figure.

Line 331: clarify

Response:

In order to satisfy your request, we have reformulated the sentence, making the text more readable.

“One-month oral N-acetylcysteine supplementation normalized glutathione levels, improved left ventricle

contractile function and ...”  

Line 384: but FIgure 5?

Response:

Figure 5 has been modified

Line 439: clarify

Response:

In order to satisfy your request, we have modified the text, line 439 now 503-505

The changes made are highlighted in yellow in the text

Reviewer 2 Report

The manuscript entitled “Dietary thiols: a potential supporting strategy in the treatment of cardiovascular and muscular disorders in sports activity” by Brancaccio et al. is a narrative review that would like to point out how healthy lifestyle habits (through a combination of moderate physical exercise and balanced diet) could prevent free radical-mediated organ impairments (mainly heart and muscles). A special focus is put on natural sulfur-containing compounds and their antioxidative properties.

Despite the importance of the issue raised by the authors, the manuscript is really disappointing and many concerns make its publication at moment questionable.

-The title and the abstract do not match properly the content of the review.

In the title the focus seems to be on the potential benefits of dietary thiols in cardiovascular and muscular disorders (intended as pathologies) in sport. On the other hand, the abstract is focused on the importance of moderate physical exercise and a diet rich in antioxidants (sulfur-containing compounds) to counteract the harmful effects of oxygen free radicals; there are no clear statements regarding “cardiovascular and muscular disorders in sport”. Then, in the review there is a section on heart failure in professional athletes and one on free radical-mediated muscle damage, but not on muscular disorders as stated in the title.

-The general reading is unpleasant, above all the first three sections (Introduction, Mechanisms of oxidation, Antioxidant role of physical exercise). These sections suffer of a highly messy structure, where many provided notions are too basic and sloppy.

-The 1st section

In the introduction (line 53) is cited the dual role of free radicals; nevertheless, the authors do not describe the “useful” one (neither in any other section of the manuscript). Moreover, there is no recall to the physical exercise, but for in the final paragraph where the aims of the review are reported.

-The 2nd section

As for the second section, I don’t agree with the sentence (lines 105-106) “a small percentage of oxygen that is not used as an energy source forms intermediate reactions known as ROS”: oxygen is not an energy source itself and the word “reactions” is a completely wrong concept here. The sentence (line 116) “ROS can damage both DNA and proteins” is a standalone statement: no other explanation is provided; while it is understandable that the focus of this review is different, this sentence needs to be commented. The sentence (line 121) “thiol groups….are essential for cysteine residues belonging to numerous key proteins such as kinases and phosphatases” contains another wrong concept (the -CH2SH side chain DEFINES the Cys amino acids!!) and another careless statement (Cys amino acids play key structural as well as functional roles in so many different classes of proteins, and kinases and phosphatases are not representative of the whole set). Again, the concept explained at lines 142-147, supported by different autocitations of the authors, is not explained with the molecular details that the reader would expect in a section called “Mechanism of oxidation”.

-The 3rd section

This is the most scarce section within the manuscript. Too many concepts left without proper references, unusual statements (Physical activity is defined as any body movement involving smooth muscle!!), confounding paragraphs (check lines 170-174 compared to line 127-129) (check lines 138-141 compared to line 116-117: are ROS able to oxidize PUFA or is lipid peroxidation able to generate ROS? The subject is complex and a clarification is needed to avoid misunderstanding). Moreover, the authors stress a lot the importance of moderate physical exercise compared to intense physical exercise: the reader would benefit from an explanation that clarify these concepts. In addition, the strong interplay between ROS production and physical exercise should be provided here.

-The 4th section

The paragraph at lines 206-210 is not clear. Please rephrase the concepts.

All the subsections related to the single sulfur-containing compounds reports the results of at least one previous study where the specific compound was tested as supplementation in athletes. The same example is not found in the Glutathione subsection: are there no cases?

In addition, are any side effects known for such supplementations?

An exhaustive table reporting the type of supplementation, the method of assumption, the type of athletes and their physical exercise program, and the results obtained in terms of physical fitness and antioxidant effects could be useful to understand the importance of the issue discussed in the manuscript.

Please check paragraph at line 272-274: it is incomplete.

-The 5th section

Here there is again a short description of the sulfur-containing compounds, described in the previous section. A more structured frame of the manuscript should be organized.

Please specify the acronym NYHA.

-The 6th section

Here a different sulfur-containing compound (methylsulfonylmethane, MSM) is discussed. Should it be included in the previous sections???

-The 7th section

In my opinion the statement at line 439 (sulfur-containing compounds represent a useful natural therapy against oxidative stress) is in contrast with the paragraph at lines 422-424.

-References

The authors adopted an odd and random system for the references reported in the manuscript: some of them are not enough appropriate (for example, references 30-32, or 54), others are completely lacking (lines 126, 129, 136, 156, 161, 164, 175-191).

-Figures

Not all figures are instrumental to the manuscript. Some of them could be merged.

Please note that Figure 1 contains two acronyms that are not specified before the recall of the figure in the text (lines 48-52) neither in the corresponding figure legend. In general figure legends are not exhaustive (above all those of figure 4 and 5)

Throughout the manuscript there are a lot of English errors and mistyping. The authors should consider a major English revision of the manuscript. A more structured frame of the manuscript should be organized, above all in the first sections.

Check also the acronyms: sometimes are used, sometimes are not, sometimes are specified again in a new section. Please verify the NAPDH acronym: at line 108 it refers to an enzyme, at line 125 to the cofactor, and at line 291 is written differently!

Author Response

(The authors gave the same response as above.)

Reviewer 3 Report

The authors provide an interesting review about the potential uses of sulphur compound supplementation during sports acitivity.

The following revisions are necessary:

Line 57: clarify

All figures: please improve graphical resolution

Line 109 and 110: delete „such as“

Line 118 and throughout: please lower-case the 2s in H2O2

Line 134: tsthe?

Line 199: „several studies“? Which ones? Were these in vitro or in vivo studies? Probably the clinical evidence for this association is insufficient?

Line 217: clarify

Line 232: should read lipoic acid

Line 257: „it“ missing. Clarify meaning of second sentence. Assume is not corrrect terminology

Line 284: „such as broccoli“

Figure 3: is there a structure for garlic? This should be clarified

Line 331: clarify

Line 384: but FIgure 5?

Line 439: clarify

Author Response

(The authors gave the same response as above.)

Round 2

Reviewer 2 Report

The authors did not produce a revised version of their manuscript with all the requested changes.

-The authors think it is right to use acronyms without introducing them appropriately. In particular, they use the acronyms ROS and RNS in the legend of figure 1 and then in the text (line 63 new version) without explaining them, while the same are then explained in lines 78 and 79. Why don't  they anticipate such explanations?

-lines 129-130 (new version): the errors were not rectified. 1) Oxygen is not an energy source, but it is used in reactions for energy production; 2) Reactive oxygen species are not intermediate reactions, but intermediate metabolites

-lines 129-130 (new version): these lines did not changed from the first version (lines 142-147 old version). The requested changes were not made.

-Statements from line 196 to 227 (new version) do not have any references, despite the numerous claims present: why?

-line 338 (new version): Vastus lateralis should be written in italic

-line 468 (new version): The differences in result ARE due to…..

-Legends of figure 4 and 5 are inadeguate as pointed out previously. Please, explain more.

Author Response

Dear Editor,

We would like to thank for the opportunity to review our manuscript entitled “Dietary thiols: a potential supporting strategy against oxidative stress in heart failure and muscular damage during sports activity”, submitted to International Journal of Environmental Research and Public Health (IJERPH).

The reviewers´ comments and suggestions were important to improve the quality of our work.

We have prepared a new version of the paper together with a point by point response attached below, regarding the reviewers´ critics and recommendations.

We hope that the revised version of the paper meets the reviewers´ expectations and that our comments clarify all questions pointed out by them.

Best regards,

Prof. Dr. Olga Scudiero

Point by point response.

Comments and Suggestions for Authors

The authors did not produce a revised version of their manuscript with all the requested changes.

Response:

Dear Reviewer, thanks for your comment.

-The authors think it is right to use acronyms without introducing them appropriately. In particular, they use the acronyms ROS and RNS in the legend of figure 1 and then in the text (line 63 new version) without explaining them, while the same are then explained in lines 78 and 79. Why don't  they anticipate such explanations?

Response:

In order to satisfy your request, we have modified the text, lines 63 and 78.

-lines 129-130 (new version): the errors were not rectified. 1) Oxygen is not an energy source, but it is used in reactions for energy production; 2) Reactive oxygen species are not intermediate reactions, but intermediate metabolites

Response:

In order to satisfy your request, we have modified the text, current lines 129-131

-lines 129-130 (new version): these lines did not changed from the first version (lines 142-147 old version). The requested changes were not made.

Response:

In order to satisfy your request, the point indicated has been revised and restructured.

-Statements from line 196 to 227 (new version) do not have any references, despite the numerous claims present: why?

Response:

In order to satisfy your request, we have added references in the point indicated.

-line 338 (new version): Vastus lateralis should be written in italic

Response:

In order to satisfy your request, we have modified the text, current line 339

-line 468 (new version): The differences in result ARE due to…..

Response:

In order to satisfy your request, we have modified the text, current line 469.

-Legends of figure 4 and 5 are inadeguate as pointed out previously. Please, explain more.

Response:

In order to satisfy you request, we have modified text, lines 448 and 493.

All the changes have been highlighted in green

This manuscript is a resubmission of an earlier submission. The following is a list of the peer review reports and author responses from that submission.